# A QUANTIFIABLE TESTING OF GLOBAL TRANSLA-TIONAL INVARIANCE IN CONVOLUTIONAL AND CAPSULE NETWORKS / CONFERENCE SUBMISSIONS

## ABSTRACT

We design simple and quantifiable testing of global translation-invariance in deep learning models trained on the MNIST dataset. Experiments on convolutional and capsules neural networks show that both models have poor performance in dealing with global translation-invariance; however, the performance improved by using data augmentation. Although the capsule network is better on the MNIST testing dataset, the convolutional neural network generally has better performance on the translation-invariance.

## 1 INTRODUCTION

Convolutional neural networks (CNN) have achieved state-of-the-art performance than the human being on many computer vision tasks Krizhevsky et al. (2012); He et al. (2015). The deep learning community trend to believe that the success of CNN mainly due to two key features in CNN, reduced computation cost with weight sharing in convolutional layers and generalization with local invariance in subsampling layers Lecun et al. (1998); Lenc & Vedaldi (2014). Due to convolutional layers are 'place-coded' equivariant and max-pooling layers are local invariant Goodfellow et al. (2016), CNN has to learn different models for different viewpoints which need big data and expensive cost.

More Generalization model should be able to train on a limited range of viewpoints and getting good performance on a much more wider range. Capsule network is robust in dealing with different viewpoints Hinton et al. (2011); Sabour et al. (2017); Hinton et al. (2018). Capsules are a group of neurons which includes the pose, colour, lighting and deformation of the visual entity. Capsule network aims for 'rate-coded' equivariance because it's the weights that code viewpoint-invariant knowledge, not the neural activities. Viewpoint changes in capsule network are linear effects on the pose matrices of the parts and the whole between different capsules layers. However, it still unclear whether capsule networks be able to generalize for global translation invariance.

Visualize and Quantify the translation-invariance in deep learning model are essential for understanding the architectural choices and helpful for developing Generalization model that is invariant to viewpoint changes. An analysis using translation-sensitivity map for MNIST digit dataset has been used to investigate translation invariance in CNN Kauderer-Abrams (2017). In this paper, we introduce a simple method to test the performance of global translation-invariance in convolutional and capsule neural network models trained on the MNIST dataset.

## 2 GLOBAL TRANSLATIONAL INVARIANT TESTING

Global translational invariance (GTI) of a deep learning model trained on the MINST data which can test by using a simple testing dataset. All images in the GTI testing dataset generated by shifting the centre of mass of a Helvetica font digit from top left to bottom right one pixel each time as shown in Figure 1. The TI testing images size is $28 \times 28$ which the same size as MNIST images. Locations of the centre of mass for each digit has 18 columns and 14 rows. In total there are $18 \times 14 \times 10 = 2520$ testing images, which cover all the possible cases of translational translations. We train all deep learning models using MNIST training dataset which includes 60000 samples, and testing on both

MNIST testing dataset with 10000 samples and GTI testing dataset with 2520 samples. Nearly all the images in the MNIST dataset are located at the centre of canvas while the GTI dataset distributes uniformly on the canvas.

Compare to translation-sensitivity maps method based on average on numerous images of each class in MNIST testing dataset and shift it to all possible translations Kauderer-Abrams (2017), our way is robust to overcome random noise due to miss labelling in the MNIST testing dataset and our GTI training dataset is much smaller. Since the GTI training dataset is identical for testing all different models, it can capture the tiny difference in those models. Another advantage of the GTI dataset is it natural to quantify the global invariance because the accuracy of model predictions on the GTI testing dataset reflects the ability of the model to dealing with global translational invariance.

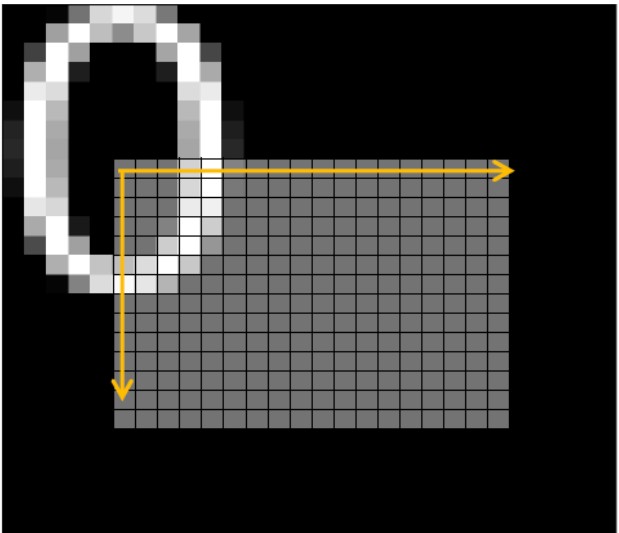

Figure 1: Global translational invariant (GTI) testing dataset. The image shows a helvetica font digit 0 located at the top left. The 18 by 14 lattice are locations of center mass of the digit if we shift the image toward right or bottom one pixel each time.

## 3 EXPERIMENTS ON GTI TESTING DATASET

### 3.1 CNN

The CNN model has nine layers as shown in Figure 2. Layer 1, 2, 4, 5 are convolutional layers with 32, 32, 64, and 64 channels respectively. Each layer has $3 \times 3$ filters and a stride of 1. Layer 3 and 6 are max-pooling layers, and layer 7, 8, 9 are fully connected layers with size 256, 128 and 10 respectively. Dropout with the drop rate of 0.5 applies to the first max-pooling layer and all the fully connected layers. The total number of parameters is 361578 which is about 23 times smaller than the Capsule networks. Except the last layer's activation function is softmax, all the other layers use ReLU. The optimizer is Adam with default parameters in Keras, and the objective function is cross entropy loss.

The results of CNN shown in Figure 3 and Table 1. The CNN models trained on MNIST data, thus it achieves very high accuracy on the MNIST testing set. However, the model trained on the MNIST training dataset without any data augmentation has only $42.16\%$ accuracy on GTI testing dataset, which implies CNN's performance on dealing with global translational invariance is abysmal. As we can see in the left picture of Figure 3, images with the digit's centre of mass around the centre of the canvas predicted correctly, and images with the number at the corner assigned to an incorrect class. Those images around the centre are classified correctly due to the max-pooling layers preserve the local invariance in feature maps. Since only MNIST images used to train the model, and the model could not accurately predict those images shift toward the corner in GTI dataset, which strongly suggests that CNN is 'place-code' equivariant.

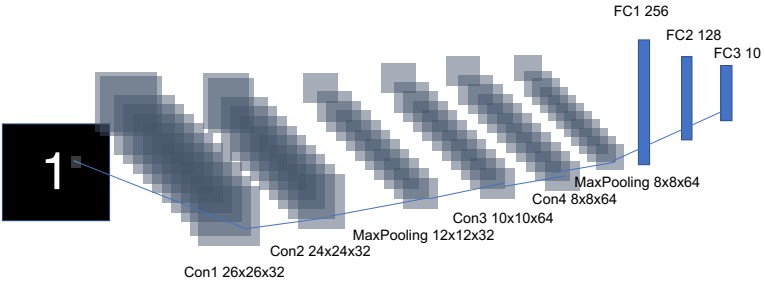

Figure 2: CNN architecture. The size and type of each layers are indicated. 'Con' refer to convolutional layers, 'FC' is fully connected layers. To save the space, not all the channels are shown in each convolutional or max-pooling layer.

Figure 3: Prediction of CNN models on digit 0's images in GTI testing set. The location of each number in the above pictures corresponding to the centre of mass lattice shown in Figure 1. (Left) The model trained on the original MNIST data without any data augmentation; (Right) the model trained on MNIST data with randomly shifting 20% width or height of the centre of mass. Correct predictions coloured by red, and black numbers are wrong predictions.

To improve the performance of CNN on the GTI dataset, we train the MNIST with data augmentation by shifting the image from the centre in x and y-direction. The accuracy on GTI testing dataset increase to 98.05% by randomly moving the centre of an MNIST training image in x and y-direction up to 30% of width or hight. Data Augmentation of training dataset to improve the performance imply place-code' equivariance in CNN, because those neurons at the corner of feature maps are activated when the model start to see training sample with objects at the edge.

## 3.2 Capsule network

We test GTI dataset on CapsNet with the same architecture in Sabour et al. (2017) bashed on the Keras Chollet (2015). The CapsNet has 8.2M parameters that are about 23 times larger than the CNN. We trained the model with Adam optimizer and using exponential decay of learning rate with decay parameter 0.9. We use margin lass with the same parameters in Sabour et al. (2017). We also add reconstruction loss but scale down it by 0.0005.

Capsule network is robust in viewpoints invariance of object recognition; however, our experiment implies capsule network's performance on global invariance has yet to be improved. As we can see

Table 1: CNN performance on MNIST and GTI testing datasets.

| MNIST error (%) | TI error (%) | Shift x (%) | Shift y (%) |
|---|---|---|---|
| $0.45_{0.01}$ | $57.84_{0.98}$ | 0 | 0 |
| $0.43_{0.06}$ | $44.38_{0.84}$ | 5 | 5 |
| $0.45_{0.03}$ | $24.63_{0.40}$ | 10 | 10 |
| $0.55_{0.04}$ | $10.05_{2.34}$ | 15 | 15 |
| $0.55_{0.07}$ | $3.81_{1.07}$ | 20 | 20 |
| $0.56_{0.06}$ | $1.95_{0.46}$ | 30 | 30 |

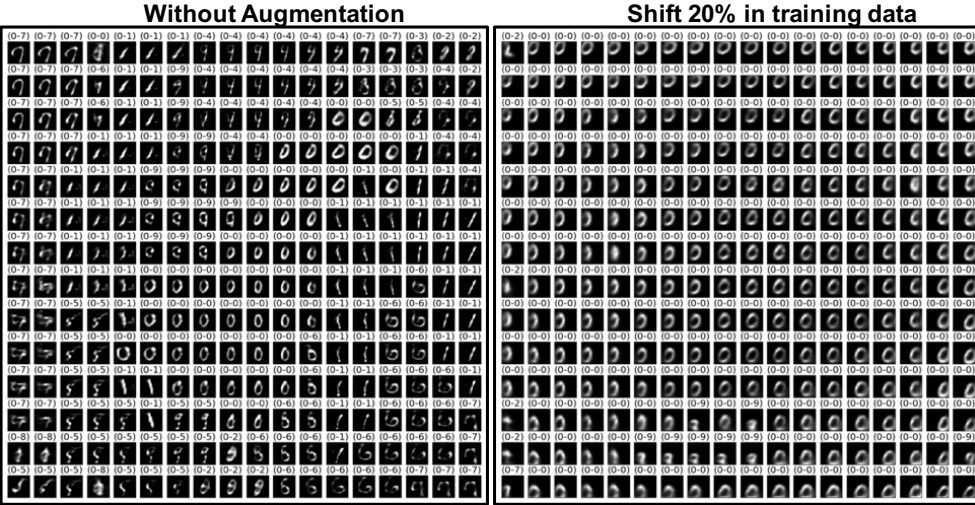

Figure 4: Generated images and prediction of CapsNet models on digit 0's images in GTI testing set. The location of each picture in the above pictures corresponding to the centre of mass lattice shown in Figure 1. (Left) The model trained on the original MNIST data without any data augmentation; (Right) the model trained on MNIST data with randomly shifting 20% width or height of the centre of mass. Inside brackets at the top of each image are ground true classes and prediction of CapsNet.

in the left of Figure 4, the model trained on the MNIST without data augmentation fails to predict the class correctly and also generated incorrect images if the digit close to the edge. Data augmentation in MNIST training dataset helps the CapsNet achieve better accuracy on the GTI dataset. The right of Figure 4 is an example trained on MNIST with 20% shifting. Nearly all images being predicted correctly except those close to the edge. It's interesting that the generated images look like handwriting even the input images are Helvetica font.

CNN's performance on GTI dataset generally better than the CapsNet as shown in Figure 5. The accuracy on CNN is always better than the CapsNet with different amount of shifting, even the convolutional layers in CapsNet using wider receptive fields. Since we remove the max-pooling layers in the CapsNet and the convolutional layers in the CapsNet is 'place-coded' equivalent, which could be the reason that the CapsNet has lower performance on the GTI dataset. We believe there is much room to improve for the CapsNet to handling translational invariance.

## 4 CONCLUSION

We introduce a simple GTI testing dataset for deep learning models trained on MNIST dataset. The goal is to get a better understanding of the ability of CNN and CapsNet to dealing with global translational invariance. Although the current version of CapsNet could not handle global translational invariance without data augmentation, we still believe CapsNet architecture potentially better than CNN on dealing with global translational invariance because capsules could train to learn all viewpoint no matter it receives the information for the centre or the edge. Our testing method is sample

Table 2: CapsNet performance on MNIST and GTI testing datasets.

| MNIST error (%) | TI error (%) | Shift x (%) | Shift y (%) |
|---|---|---|---|
| $0.47_{0.02}$ | $67.46_{0.54}$ | 0 | 0 |
| $0.43_{0.03}$ | $53.41_{0.62}$ | 5 | 5 |
| $0.36_{0.02}$ | $32.54_{0.35}$ | 10 | 10 |
| $0.39_{0.04}$ | $16.07_{1.12}$ | 15 | 15 |
| $0.50_{0.06}$ | $7.62_{1.21}$ | 20 | 20 |
| $0.75_{0.08}$ | $6.55_{0.69}$ | 30 | 30 |

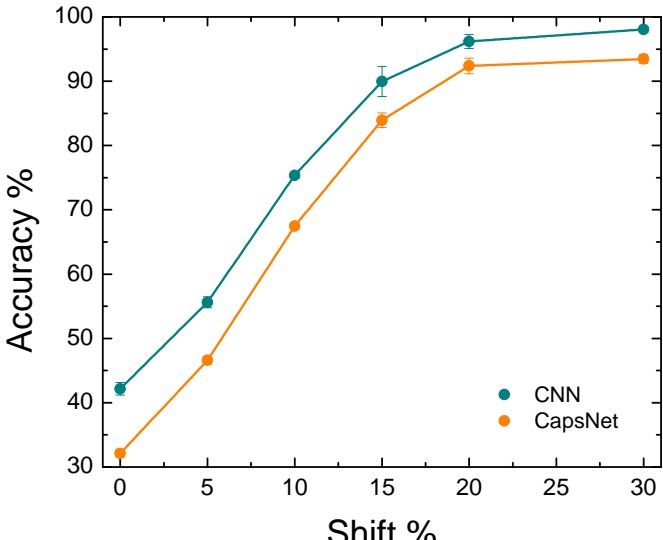

Figure 5: GTI dataset accuracy of models trained on CNN and CapsNet with different amount of random shifting in MNIST training dataset.

and quantifiable, and it easy to implement for other datasets of computer vision tasks by taking a clear and correct labelled image from each class and apply the translational shifting to cover all possible cases.

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
