# OpenReview forum: "A quantifiable testing of global translational invariance in Convolutional and Capsule Networks"
_ICLR.cc/2019/Conference_

### Official Review · AnonReviewer3 · 2018-10-25
**A simple test of CNN and CapsNet's performance on translation**

**Rating:** 3
**Confidence:** 5

**Review:**

The authors of this paper compare the robustness of CNN and CapsNet to global translation on the MNIST dataset. Both models were trained on the standard training set of MNIST, and then tested on a set with digits shifting from the upper left corner to the lower right corner. The results of both models were poor. To improve it, the authors add some shifted digits to the training set, and the performances of both models were significantly enhanced. Moreover, the performance of CNN was better than that of CapsNet in the experiments. Generally speaking, the work presented in this paper is clear and straightforward. However, the work is not significant enough to publish as a ICLR paper. Below is my major comments.

1. There are lots of typos and grammatical errors everywhere in the paper. Thus, the manuscript was not well prepared.

2. It is unclear which CapsNet and what settings were used in the experiment.

3. It is well-known that convolutional networks are good at capturing local patterns from the images, while capsule networks enhance it to consider global configurations of the local patterns, and robust to affine transformation. Obviously, the experiments presented in this manuscript is too simple. Lots of work should be done in the investigation. For example,o on the training set and shifted test set, the authors can enlarge the background and keep the digits in the original size to make it as a local pattern in the image. Will it be detected by CNNs with larger receptive fields for the images? How is it compared with CapsNets?

4. How are both models compared on other (perhaps more complicated and larger) datasets?

In summary, the work presented here is interesting, but lots of work should be done in order to make it publishable.

---

### Official Review · AnonReviewer2 · 2018-10-28
**Clever idea with some weaknesses...**

**Rating:** 4
**Confidence:** 4

**Review:**

Thanks for the submission of you work. As far as I understood it correctly you deal with the idea to test the shift invariance of a given model on Helvetica digits. You propose that idea as general quantifiable approach.

In general, your paper is well-written, compact, in a good style and with a length of 5 pages really short.

In the introduction, you cite the work of Hinton and Sabour and describe the Capsule framework in general. I think that this description is not really precise. The aforementioned contributions are really different and your general explanation is technically incorrect.

The major concerns about your contributions are:
1. Helvetica digits are not in American digit style, e.g. check the digit one. The MNIST database consists of American handwritten digits. Why you’re not using an American digit font? What is the impact on your model due to that change?
2. Fig. 1 in your contribution: It seems that your digits are too small compared to MNIST digits. Is that true? Note that MNIST digits are size normalized.
3. What is the outstanding advantage of your proposal compared to a simple shift of a MNIST digit or the usage of the affNIST dataset?
4. Page 4: You are mentioning that it is interesting that the reconstructed image looks like a handwritten digit. Why you think it is interesting? I would assume that this is a natural behavior since your network was trained to do so.

Please clarify the questions above and highlight what is the advantage of your method compared to simple shifts of MNIST digits or even the usage of affNIST. Right now, I’m not seeing a real advantage neither a scientific contribution.

---

### Official Review · AnonReviewer1 · 2018-11-01
**No technical novelty or implementation described**

**Rating:** 3
**Confidence:** 5

**Review:**

Authors present a study to compare global translation invariance capabilities of CNNs and CapsuleNets.
The paper doesn't introduce any novel concept or technique but it simply compares two established techniques on MNIST dataset. The interest on this paper is rather limited. Besides many technical concepts are not really accurate on how they are presented. It needs further attention and improvements.
The paper reads more like a review papers than a new research article. I remain to my initial decision.

---

### Meta-Review · Area_Chair1 · 2018-12-04
**decision**

**Confidence:** 5
**Recommendation:** Reject

**Metareview:**

The paper presents an empirical comparison of translation invariance property in CNN and capsule networks. As the reviewers point out, the paper is not acceptable quality at ICLR due to low novelty and significance.